# Practical Tips for Safe and Successful Endoscopic Ultrasound-Guided Hepaticogastrostomy: A State-of-the-Art Technical Review

**DOI:** 10.3390/jcm11061591

**Published:** 2022-03-14

**Authors:** Saburo Matsubara, Keito Nakagawa, Kentaro Suda, Takeshi Otsuka, Masashi Oka, Sumiko Nagoshi

**Affiliations:** Department of Gastroenterology and Hepatology, Saitama Medical Center, Saitama Medical University, 1981, Kamoda, Kawagoe 350-8550, Japan; kate-ill@hotmail.co.jp (K.N.); leclearlshelly@gmail.com (K.S.); ohitoyosinokaze@yahoo.co.jp (T.O.); oka@dd.iij4u.or.jp (M.O.); snagoshi@saitama-med.ac.jp (S.N.)

**Keywords:** hepaticogastrostomy, endoscopic ultrasound, endoscopic ultrasound-guided biliary drainage (EUS-BD), endoscopic ultrasound-guided hepaticogastrostomy (EUS-HGS)

## Abstract

Currently, endoscopic ultrasound-guided hepaticogastrostomy (EUS-HGS) is widely performed worldwide for various benign and malignant biliary diseases in cases of difficult or unsuccessful endoscopic transpapillary cholangiopancreatography (ERCP). Furthermore, its applicability as primary drainage has also been reported. Although recent advances in EUS systems and equipment have made EUS-HGS easier and safer, the risk of serious adverse events such as bile leak and stent migration still exists. Physicians and assistants need not only sufficient skills and experience in ERCP-related procedures and basic EUS-related procedures such as fine needle aspiration and pancreatic fluid collection drainage, but also knowledge and techniques specific to EUS-HGS. This technical review mainly focuses on EUS-HGS with self-expandable metal stents for unresectable malignant biliary obstruction and presents the latest and detailed tips for safe and successful performance of the technique.

## 1. Introduction

Endoscopic ultrasound-guided biliary drainage (EUS-BD) has become a promising alternative to percutaneous transhepatic biliary drainage (PTBD) after difficult or failed endoscopic retrograde cholangiopancreatography (ERCP) in patients with benign or malignant biliary obstruction [1,2,3,4]. Furthermore, its applicability as a primary drainage has also been reported [5,6,7]. The technique of EUS-BD is divided into rendezvous with ERCP, antegrade stenting, and bilioenterostomy, which includes EUS-guided hepaticogastrostomy (EUS-HGS) and EUS-guided choledochoduodenostomy (EUS-CDS) [8]. Among these techniques, EUS-HGS has the broadest indications, including duodenal stenosis [9], surgically altered anatomy [10], high-grade hilar stenosis [11,12] as well as failed biliary cannulation, and is therefore considered to be the most frequently performed technique in EUS-BD [13,14]. However, EUS-HGS can cause serious adverse events such as bleeding [15], bile leak leading to peritonitis or biloma/abscess, perforation, focal cholangitis, and stent migration [16].

Several guidelines or technical reviews on EUS-HGS have been reported [8,17,18]. However, techniques and devices are constantly evolving, and it is necessary to keep up to date with the latest advances. This latest technical review provides detailed tips and tricks for safe and successful EUS-HGS using many easy-to-understand illustrations and figures with reference to the recent literature.

## 2. Physician and Facility Requirements

EUS-HGS is a technically complex procedure with life-threatening risks and should be performed by a physician with extensive experience and skill in ERCP and basic EUS-guided procedures such as fine needle aspiration (FNA) and peripancreatic fluid collection drainage [8]. If a physician is performing EUS-HGS for the first time, the procedure should be performed under the supervision of an expert with adequate experience in EUS-HGS. Physicians and assistants must be familiar with endoscopic system and various accessories including FNA needles, guidewires, dilation devices, and stents. Furthermore, it is important that immediate support from interventional radiologists and surgeons are available in case of serious adverse events such as arterial bleeding or migration of the stent into the abdominal cavity [8].

## 3. Preparation for a Safe Procedure

Contrast-enhanced computed tomography (CT) prior to EUS-HGS is essential to evaluate not only the biliary tree but also ascites, collateral vessels, tumor location in the liver, and distance between the left hepatic lobe and the lesser curvature of the stomach. If ascites is present between the left hepatic lobe and the lesser curvature of the stomach, a fistula will not form after EUS-HGS, and even if a covered self-expandable metal stent (SEMS) is used, bile, gastric juice, and air may leak into the abdominal cavity over time, causing peritonitis. Therefore, EUS-HGS should not be performed in patients with uncontrollable ascites in this region [19,20]. Collateral vessels are often observed around the stomach due to tumor invasion of the portal vein or splenic vein. In such cases, the feasibility of EUS-HGS is not known until EUS observation is performed in Doppler mode, so an alternative drainage plan should be prepared before starting the procedure. Likewise, if there is a tumor in segment 2 or 3 of the liver, a backup plan should be discussed beforehand, as it is not known whether EUS-HGS can be carried out while avoiding the tumor until EUS observation is conducted. A long distance between the liver and stomach before EUS-HGS may increase the risk of migration of the gastric end of the stent into the abdominal cavity after EUS-HGS [21], so it is advisable to use a stent of sufficient length or a stent with an anti-migration system.

## 4. EUS System

The EUS system is comprised of an echoendoscope with a curved linear array transducer and a processor. Optically, the oblique-viewing echoendoscope is the most common type in ES-HGS, while some endoscopists prefer the forward-viewing type [22]. There are three types of EUS systems available worldwide (Table 1). EG-580UT (Fujifilm Medical Corp, Tokyo, Japan), GF-UCT260 (Olympus Medical Systems, Tokyo, Japan), and EG38-J10UT (Pentax medical, Tokyo, Japan) are oblique-viewing echoendoscopes that have large bore accessory channels. EG-580UT and EG38-J10UT have better maneuverability with greater vertical angle mobility than GF-UCT260. Meanwhile, GF-UCT260 has a greater range of the ultrasound view, which helps to identify intervening mucosa or vessels before advancing the needle into the gastric wall. EG-580UT and GF-UCT260 have dedicated ultrasound processors (SU-1; Fujifilm Medical Corp, EU-ME2; Olympus Medical Systems), which can be mounted on an endoscope trolley and have ancillary functions: Doppler mode and contrast harmonic mode. The former allows the needle to avoid vessels when puncturing. Contrast-enhanced EUS using the latter function facilitates the identification of bile ducts when they are obscured by echogenic lesions such as stones or sludge [23]. EG38-J10UT does not have a dedicated processor, so it needs to be connected to an external ultrasound platform (ARIETTA series; Fujifilm Medical Corp).

## 5. Step-by-Step Tutorial on EUS-HGS Procedure including Devise Selection

In EUS-HGS, the left lateral branch of the intrahepatic bile duct is first punctured from the stomach or jejunum (in the case of post-gastrectomy) with an FNA needle, followed by injection of contrast medium and insertion of a guidewire. After the needle is removed, a dilation device is inserted into the bile duct to dilate the tract. Next, an introducer of a SEMS or plastic stent (PS) is inserted into the bile duct. Finally, a stent is deployed between the bile duct and the stomach or jejunum (Figure 1).

### 5.1. Selection of Bile Duct Puncture Site and Scope Position

The intrahepatic bile ducts (B2 or B3) in the left lateral lobe of the liver are candidates for the puncture. On EUS imaging, B2 is directed from the B2/B3 junction to the right superiorly, and B3 is directed to the left superiorly [24]. Therefore, the B2 puncture is easier for inserting the guidewire into the bile duct because the trajectory of the needle and the direction of the bile duct are similar. However, most experts prefer to puncture B3 rather than B2 because puncturing B2 can be a transesophageal puncture, which may result in the risk of mediastinitis [8,25]. Because the position of the segment 2 of the liver is more cephalad than the segment 3, the position of the scope when puncturing B2 is shallower than that of B3, and even if the transducer is in the stomach, the exit of the accessory channel is often in the esophagus.

Before starting B3 puncture, it is desirable to adjust the position of the scope and the direction of the needle. For easy and reliable manipulation of the guidewire toward the hilum, the angle formed by the needle and the bile duct on the hilar side should be obtuse. When the scope is in a shallow position, that angle is often acute, making it difficult to manipulate the guidewire toward the hilum (Figure 2A,B); pushing the scope while turning the large wheel upward rotates the EUS image clockwise and makes that angle obtuse (Figure 3A,B). In fact, Ogura et al. reported in a retrospective multivariate analysis that strongly applying the up-angle of the scope to make the angle between the scope and the needle less than 135 degrees was a positive predictive factor of successful guidewire manipulation toward the hilum [26]. However, this bent scope shape reduces the forward push force during device insertion, and in the worst case, the scope may be pushed back, and the guidewire may be dislodged from the bile duct. Shiomi et al. [27] and Nakai et al. [28] reported the usefulness of the “Double guidewire technique” using a double lumen catheter (Uneven Double Lumen Cannula [UDLC]; Piolax Medical Device, Kanagawa, Japan), which allows a second 0.035 inch guidewire to be inserted adjacent to the first 0.025 inch guidewire (Figure 4). This technique improves the stability of the scope during device insertion and allows the use of the stiffer second guidewire if necessary. In addition, the second guidewire can be used to perform another stent insertion in case of a failed stent insertion, ensuring a safe procedure.

The choice of puncture site is important; Oh et al. reported that a bile duct diameter >5 mm and a distance ≤3 cm from the hepatic surface to the punctured bile duct at the puncture site were associated with technical success [29]. On the other hand, Yamamoto et al. reported that bile peritonitis was more likely to occur when the distance between the hepatic surface and the punctured bile duct was less than 2.5 cm [30]. Taking these factors into consideration, we believe that puncture at B3 close to the B2/3 bifurcation is the best choice. This is because the bile duct diameter is large, which makes puncture easy; the liver parenchyma is sufficiently intervened to avoid bile leakage; and the angle between the needle and the bile duct on the hilar side is obtuse, which facilitates successful insertion of the guidewire into the hilar bile duct (Figure 3A,B). If the biliary stricture is close to the B2/3 bifurcation, the puncture point must be on the peripheral side in order to secure the space in the bile duct for stent placement.

When performing a B2 puncture, it is of paramount importance to avoid transesophageal puncture. There are several methods to achieve this, such as confirming the needle puncture position under direct endoscopic view, clipping the esophagogastric junction and confirming it under fluoroscopy [25], or confirming the diaphragmatic crus by ultrasound. If the scope is shallow, the needle and B2 are parallel to each other, making puncture difficult and increasing the risk of transesophageal puncture (Figure 5A,B); therefore, slightly pushing the scope while turning the large wheel upward facilitates transgastric and reliable bile duct puncture (Figure 6A,B).

### 5.2. Biliary Puncture

There are various types of FNA needles, each with a different tip shape and different materials for the needle and sheath. Nitinol needles are more flexible and less prone to bending than steel needles. Additionally, the coil sheath has a higher lumen retention when bent than the plastic sheath. These properties are useful for performing EUS-HGS. The EZ-shot 3 plus (Olympus Medical Systems) (Figure 7A) is the only commercially available nitinol needle with a coil sheath. In EUS-HGS, one of the most difficult steps is the manipulation of the guidewire through the needle [31]. The main issue is guidewire shearing, which in turn created a risk of leaving a tip of the guidewire in the patient. The EchoTip Access Needle (Cook Medical, Winston Salem, NC, USA) is a dedicated needle for interventional EUS, which has a sharp stylet for puncture, and the needle tip becomes blunt when the stylet is removed, thus avoiding guidewire shearing [17,32] (Figure 7B).

As for the needle size, a 19-gauge needle is preferable to a 22-gauge needle because a 0.025 inch guidewire can be used, which performs better than a 0.018 inch guidewire. Usually, a 22-gauge needle is used with a 0.018 inch guidewire for thin bile ducts.

Prior to inserting the needle into the accessory channel of the scope, remove the biopsy valve from its socket and attach it to a dilation device (Figure 8A). Before puncture, remove the stylet of the needle and place a syringe filled with contrast medium to pre-fill the lumen with contrast medium (Figure 8B).

Unlike PTBD, in EUS-HGS, the scope moves with the liver and stomach due to respiration, and thus the fluctuations of the liver on the ultrasound image are small. Therefore, rapid puncture is usually not necessary, and careful puncture is advisable to avoid intervening vessels. However, if the bile duct wall is stiffened due to fibrosis (due to prior biliary drainage or cholangitis), a slow puncture speed will not allow the needle to be inserted into the bile duct. In such cases, the needle should be punctured quickly and strongly, once penetrating the bile duct wall completely. After penetration, the needle is slowly withdrawn while applying suction pressure (Seldinger method) [33,34]. The success of the bile duct puncture is confirmed by aspiration of bile usually, but it is not possible to aspirate bile if the bile duct is narrow. In such cases, when the needle enters the bile duct, the air drawn from the stomach by the aspiration enters the bile duct and is recognized as a moving strong echo. This is a useful finding to determine the success of the puncture.

If a favorable biliary puncture line cannot be obtained due to the intervening vessels or tumors, or due to the alignment of the liver and stomach, pressing the scope after advancing the needle into the liver parenchyma can move the liver to the right and rotate it counterclockwise on ultrasound image using the liver access point as a fulcrum, thereby can alter the trajectory of the needle (Figure 9A,B). Ishiwatari et al. also reported the “Bent needle technique” in which a manually pre-bent needle is used to puncture in such a case [35].

### 5.3. Contrast Injection

If contrast medium is injected directly after bile duct access, the intraductal pressure will increase. The increased intraductal pressure may not only cause bile leak but also cause cholangio-venous reflux, which may lead to bacteremia in case of cholangitis. Therefore, it is necessary to aspirate as much bile as possible before injecting the contrast medium. Ishiwatari et al. reported in a retrospective study that bile aspiration of 10 mL or more was a significant factor in reducing the occurrence of adverse events associated with bile leak [36]. In this study, a catheter was inserted into the bile duct to aspirate bile prior to tract dilation, which requires more steps in the procedure; therefore, bile aspiration with an FNA needle seems preferable. Following bile aspiration, contrast medium is injected to depict the biliary tract. In order to improve the handling of the guidewire through the needle and the visibility of the guidewire under fluoroscopy, it is recommended to use a contrast medium diluted to half its concentration in saline. The amount of contrast medium injected should be limited to the minimum amount that will allow the hilar region to be visualized to avoid increased intraductal pressure.

### 5.4. Guidewire Manipulation

When using a 19-gauge needle, a 0.035 inch or 0.025 inch guidewire can be used. However, the 0.025 inch guidewire is preferable because there is less risk of the guidewire being sheared by the needle tip and it is easier to manipulate. In recent years, a number of 0.025 inch guidewires have been released, such as VisiGlide2 (Olympus Medical Systems), EndoSelector (Boston Scientific Corp, Natick, MA, USA), M-Through (Medicos Hirata, Osaka, Japan), and INAZUMA (Kaneka Medix, Osaka, Japan), which have a hydrophilic coating on the tip, a stiff shaft, and excellent torque and supportability. When using a 22-gauge needle, a 0.021 inch or 0.018 inch guidewire can be used, but the performance of these conventional guidewires has not been sufficient. Most recently, a new 0.018 inch guidewire (Fielder 18; Olympus Medical Systems) has been released, which has a high performance similar to that of the 0.025 inch guidewire [22,37,38].

The guidewire is advanced through the needle, and once it enters the bile duct, it is slowly and carefully advanced with gentle rotation to guide it toward the hilar region. If the guidewire is unintentionally advanced to the peripheral side, the “Loop technique” should be attempted first. Push the guidewire with rotation, and when the tip of the guidewire is caught on a lateral branch (Figure 10A), push the guidewire further. Since the tip of the guidewire is fixed, the body of the guidewire will bend with the pushing force and form a loop (Figure 10B). If the loop is facing the hilar region, the guidewire can be advanced to the hilum by pushing further (Figure 10C,D). If the “Loop technique” fails, the “Moving scope technique” is an alternative to change the direction of the guidewire, where pushing the scope while turning the large wheel upward may change the direction of the needle to the cranial side, allowing the guidewire to proceed toward the hilum [39] (Figure 11A–C).

When these methods are unsuccessful, the guidewire must be pulled out and reoriented toward the hilar region. However, if there is any resistance while pulling, the guidewire should not be pulled out forcibly because the tip of the guidewire may be sheared off and remain as a foreign body. In such a case, it is recommended to pull the guidewire out while slowly moving it back and forth with rotation. If this does not work, Ogura et al. reported the usefulness of the “Liver impaction technique” [40]. By pulling the needle tip slightly into the hepatic parenchyma, the angle between the guidewire and the needle is loosened, and the tip of the needle is covered by the hepatic parenchyma to prevent shearing the guidewire (Figure 12A–D). The aforementioned dedicated needle (EchoTip Access Needle; Cook Medical) is expected to prevent shearing of the guidewire [17,32], but is not yet widely available in the world.

If changing the direction of the guidewire is not successful even using these techniques, it is necessary to change the needle to a catheter to improve the manipulation of the guidewire. However, the guidewire and catheter may become dislodged from the bile duct while struggling to change the direction of the guidewire by pulling the tip of the catheter back to the shallowest part of the bile duct. In order to avoid such an eventuality, the aforementioned “Double guidewire technique” using UDLC is effective. While securing the bile duct with the first guidewire, the second guidewire is manipulated to advance to the hilar region [41]. Although UDLC is a double-lumen catheter, the second guidewire is located away from the tip, allowing the tip to be thin enough to be inserted directly into the bile duct without pre-dilation.

In cases where the guidewire cannot be redirected using UDLC, a rotatable sphincterotome (TRUEtome; Boston Scientific Corp) may be of assistance. After two guidewires are implanted in the peripheral bile duct and removal of UDLC, TRUEtome is inserted into the bile duct over the guidewire. Then, the guidewire is directed to the hilar region by rotating and bending the tip of TRUEtome while securing the bile duct with another guidewire (Figure 13A–C) [42].

If all else fails, the only option is to withdraw the needle completely and re-puncture the bile duct. The recently developed “steerable access device”, which has a bendable needle tip, allows the guidewire to direct the hilar region easily and reliably [43,44]. However, this device has not yet been made widely commercially available in the world.

Once the guidewire has passed through the stricture, it must remain in place as long as possible to prevent dislodgement by the assistant’s pulling during subsequent insertion of the device.

### 5.5. Tract Dilation

After a sufficient length of guidewire is placed, the needle is replaced with a dilatation device. In ERCP, the elevator is usually raised completely after device removal to prevent guidewire dislodgement. However, in EUS-HGS, the elevator should not be raised further after the needle is removed, because it is most critical to maintain ultrasound visualization of the puncture line to ensure subsequent device insertion. The more skilled the physician is in ERCP, the more likely it is that he or she will do this unconsciously, so care must be taken.

The dilatation of the tract is carried out using a mechanical dilator such as a bougie dilator or balloon dilator, or a diathermic dilator. The bougie dilator is the safest, but insertion of an introducer of covered SEMS is often difficult because the size of the hole opened on the bile duct is the smallest, usually only 7 Fr. The balloon dilator can make the largest hole, but it is associated with the risk of bile leak. The diathermic dilator is the most reliable in penetrating the bile duct wall, but the burning effect can cause bleeding from the surrounding liver parenchyma and hepatic artery. Therefore, the bougie dilator is appropriate for stents with small caliber introducers (7 Fr or less), such as plastic stents and some kinds of covered SEMS, while the balloon dilator is suitable for conventional covered SEMS where the introducer is usually 8 Fr or more. The diathermic dilator had better be used as a rescue when the bile duct wall is too hard to be breached by other dilators [8].

In the initial era of EUS-HGS, mechanical dilation was accomplished gradually: the ERCP catheter was inserted first after the needle removal, followed by sequential dilatation with a bougie dilator or balloon dilator [45,46,47]. Recently, however, the properties of mechanical dilators have been improved so that they can be inserted directly without dilation by the ERCP catheter. Balloon dilators include Hurricane RX (Boston Scientific Corp), which has a rigid shaft with a stylet (Figure 14A) [46], and REN (Kaneka Medics), which has an ultra-thin tip of 3 Fr (Figure 14B) [48]. ES dilator (Zeon Medical, Tokyo, Japan) is a 7 Fr bougie dilator which has an ultra-thin tip of 2.5 Fr (Figure 14C) [49,50,51]. REN and ES dilator are dedicated dilation devices for EUS-HGS that are adapted to 0.025 inch guidewires, and the gap between the tip of these devices and the 0.025 inch guidewire is extremely small.

Balloon dilation is usually performed with a 4 mm or 3 mm diameter balloon, which creates a larger tract than a bougie dilator or diathermic dilator and is therefore more prone to bile leak. The “Segmental dilation method” may be beneficial in preventing bile leak. As previously stated, it has been reported that a short distance of intervening liver parenchyma (≤2.5 cm) is more likely to cause biliary peritonitis, and in this study, balloon dilation was performed in in around two-thirds of cases [30]. Usually, balloon dilation is performed by first dilating the bile duct wall and then the gastric wall, but since the balloon is as long as 4 cm or 3 cm, the dilated portions on both sides partially overlap each other, creating a thick path from the bile duct to the extrahepatic area, causing bile to flow out. This phenomenon is especially likely to occur when the distance of the hepatic parenchyma is short and is thought to be one of the reasons for the results of the aforementioned study that biliary peritonitis is more likely to occur when the distance of the hepatic parenchyma is short. To avoid this phenomenon, the balloon catheter should be pushed into the bile duct as deeply as possible when dilating the bile duct wall and pulled into the scope channel as long as possible when dilating the gastric wall to prevent overlap of the two dilated areas (Figure 15A,B). The hepatic parenchyma left un-dilated is thought to prevent bile leakage due to the tamponade effect.

Regarding the diathermic dilation, a wire-guided needle knife was initially used in EUS-HGS. Although this type of catheter could be advanced over the guidewire, the axis of the needle was misaligned with the guidewire at the site of bending, which could cause bleeding from the surrounding organs [52,53]. In fact, Park et al. reported that the use of the needle knife was significantly associated with post-procedure adverse events compared to gradual dilation using a mechanical dilator [52]. To address this major concern, a fine diameter (6 Fr) coaxial diathermic dilator (Cysto-Gastro set; Endo-flex, Voerde, Düsseldorf, Germany) was developed to allow for safer dilation [53]. However, even with this coaxial dilator, the risk of bleeding appears to be higher than with mechanical dilators [50]. Honjo et al. reported that in EUS-HGS, bleeding occurred in 5/23 (21.7%) patients with 6 Fr Cysto-Gastro set and 0/26 patients with the bougie dilator (*p* = 0.04) [50]. Since all bleeding cases used plastic stents and spontaneous hemostasis was achieved with conservative therapy alone without interventional radiology (IVR), the bleeding was not arterial but from the surrounding hepatic parenchyma due to the burning effect. Recently, Ogura et al. reported a pilot study using a new coaxial diathermic dilator (Fine025; Medicos Hirata) with a smaller diathermic ring at the tip and less burning effect on the surrounding tissues than the Cysto-Gastro set [54]. In this pilot study, 12 patients had no adverse events. Since this dilator has a thinner tip and thicker shaft than Cysto-Gastro set, it does not need to cauterize the liver parenchyma and only needs to cauterize the gastric and bile duct walls, which may reduce bleeding. However, since the burning effect on the surrounding tissues cannot be completely eliminated, arterial bleeding might be caused from the interlobular artery in the Glisson’s sheath when cauterizing the bile duct wall.

### 5.6. Stent Deployment

In the early days of EUS-HGS, plastic stents were predominantly used [55,56,57,58]. Although plastic stents are inexpensive and easy to place, they are prone to stent clogging due to their small caliber and bile leakage due to their lack of self-expandability. Therefore, conventional biliary-covered SEMS with a length of 6 cm or 8 cm have come into use in the expectation of preventing bile leaks by closing the fistula with self-expandability and prolonging the stent patency period with a large diameter [59,60]. In fact, the adverse events of EUS-HGS with a covered SEMS have been reported to be lower than with a plastic stent [53]. However, the migration of the gastric end of the stent into the abdominal cavity leading to fatal biliary peritonitis has been recognized as a major problem with a covered SEMS. For this reason, some experts initially recommended the use of a plastic stent for EUS-HGS and its replacement with a covered SEMS after fistula maturation [61,62]. However, recent advances in methodology and instrumentation have made it possible to prevent migration.

Migration can occur in two situations: early migration, when the stent detaches from the scope [63,64,65,66], and delayed migration, after successful deployment [16,62,67,68,69,70]. In EUS-HGS, the stomach and liver are initially brought in closer together by pushing the echoendoscope against the gastric wall. However, the distance between the liver and stomach becomes increased because the scope must be moved away from the gastric wall to eventually release the stent. This event and the shortening of the SEMS can cause early migration, in which the gastric end of the SEMS is pulled into the abdominal cavity. Recently, early migration can be avoided by using the “Intra-channel (conduit) release method” (see below), which can ensure that the end of the SEMS is placed in the stomach while minimizing the distance between the liver and stomach. However, since the stomach will eventually return to its original position, delayed migration may occur if the initial distance between the liver and stomach is long [21]. To prevent delayed migration, a long (≥10 cm) SEMS is recommended to ensure sufficient intragastric stent length [8,71,72]. Nakai et al. [71] and Ogura et al. [72] reported that sufficient intragastric length (>30–35 mm on CT the next day) may not only prevent delayed migration but also prolong stent patency by reducing the reflux of gastric juice and food. Nevertheless, even in cases with long intragastric stent length, the stent may be migrated by sudden gastric movements such as hiccups or vomiting [71]. Therefore, long stents with anti-migration properties may be optimal [21].

Currently, various types of SEMS are available for EUS-HGS with respect to stent design (braided or laser-cut type), coverage (partial or full), presence or absence of anti-migration properties at the gastric end, and size of the introducer. As a dedicated device for EUS-HGS, several partially covered braided SEMSs with anti-migration properties have been released by Korean companies (Figure 16A–D) [17,73,74,75,76,77,78,79]. In Japan, the most common SEMS for EUS-HGS is Niti-S S-type stent (modified Giobor stent; Taewoong Medical, Seoul, Korea), which is a partially covered SEMS with a 1 cm uncovered portion at the hepatic end [71,80]. Since this stent is a braided SEMS with a cross-wire structure, it gradually expands in the stomach from the non-expanded part in the gastric wall to form a smooth and gently sloping stent surface. Therefore, the effect of holding down the gastric wall is weak. Furthermore, the shortening rate of the stent is large, which tends to cause delayed migration of the gastric end into the peritoneal cavity (Figure 17A–C). In order to prevent this, the stent length should be longer than 10 cm, but even a long stent cannot prevent it completely as mentioned above. For this reason, Niti-S Spring Stopper Stent (Taewoong Medical) was developed with a spring-type stopper at the gastric end to prevent migration (Figure 18). This stent can reliably prevent delayed migration of the gastric end. Meanwhile, pre-dilation of the tract is usually required for these SEMSs insertion because the diameter of the introducer is 8.5 Fr.

There are several SEMSs with a slim introducer allowing direct insertion without prior tract dilation. From Korea, HANAROSTENT Benefit (M.I.Tech, Seoul, Korea) [22,81,82] and EGIS Braided 6 (S&G Biotech, Seongnam, Korea) [83], which are fully covered SEMSs with a 6 Fr introducer for a 0.025 inch guidewire, have been released. In most cases, these SEMS can be inserted without prior dilation. Nevertheless, since these SEMS are of the fully covered type without any anti-migration properties, migration of both sides is feared. In addition, since the bile ducts on the peripheral side of the access point are dead spaces, these SEMS are not only unsuitable for hilar biliary obstruction but may also cause focal cholangitis in the dead spaces [84]. Most recently, Covered BileRush Advance (Piolax Medical Devices), a partially covered SEMS with a 2 cm uncovered portion at the hepatic end, has been launched (Figure 19A). This stent has an introducer compatible with a 0.025 inch guidewire that has a 2.4 Fr tip and a 7 Fr shaft and can be inserted directly without dilation in most cases (Figure 19B). Because this stent is a laser-cut type, the stent expands rapidly in the stomach from a non-expanded area in the gastric wall, resulting in a steep stent surface. This incised shape and jagged struts inhibit gastric wall return to its original position (Figure 20A,B); furthermore, there is almost no shortening of the stent, which results in little delayed migration [85]. One-step EUS-HGS without prior tract dilation has the potential to reduce adverse events and procedure time compared to conventional methods, and further studies are warranted.

The process of partially covered SEMS deployment is as follows. First, proper positioning of the stent introducer is performed. When the introducer is inserted into the bile duct, the scope is pushed back by the counteraction and the distance between the liver and stomach is increased. Stent deployment must not be started at this point, as the stent end may fall into the abdominal cavity. The introducer should be inserted deeply once and then pulled to adjust its position so that only the uncovered portion enters the bile duct. This pulling motion will shorten the distance between the liver and stomach. The next step is to detach the SEMS from the introducer. After positioning the introducer, the assistant pulls on the outer sheath to gradually release the SEMS. At this time, the introducer is retracted into the scope channel due to the counteraction, and the stent is advanced. The physician must pull the introducer as the assistant works, while watching the fluoroscopic view to ensure that the tip position of the stent remains the same. Once the uncovered area is fully expanded, the stent is fixed to the liver, and the physician’s pulling force draws the liver into the scope, bringing the liver and stomach even closer together. The last step is SEMS implantation, which requires the scope to be pulled away from the gastric wall in order to bring the SEMS out of the channel. If the scope is simply pulled back, the pushing force of the scope will be lost, and the gastric wall will be moved away from the liver. As a result, the stent length in the abdominal cavity becomes longer while the stent length in the stomach becomes shorter, and the end of the stent may migrate into the abdominal cavity. To avoid this problem, “Intra-channel (conduit) release method [86,87]” is essential. The physician pulls the introducer as the assistant moves to deploy the stent but stops the deployment once the fluoroscopy shows that the tip of the outer sheath has been pulled about 1 to 2 cm inside the channel. At this point, the physician pushes the introducer in the opposite direction, and the expanded portion of the stent emerging from the channel is pressed strongly against the gastric wall (Figure 21A). This action creates a gap between the scope and the gastric wall, and stent deployment across the gastric wall can be directly confirmed (Figure 21B). Afterwards, the assistant resumes pulling the outer sheath, and the released stent pushes the gastric wall forward, and the counteraction pushes the scope back. By gradually loosening the push of the introducer and the up angle of the scope while feeling the counteraction force, the scope can be released from the gastric wall while keeping the stomach and liver close together, and finally the stent is completely released in the stomach. The trick of this method is to push the expanded part of the stent, which has been partially released in the channel, against the gastric wall; pushing without intra-channel release will only cause the introducer to enter the fistula. 

## 6. Post-Procedure Management

If abnormal findings are found on laboratory tests or physical examination the day after the procedure, or if sufficient intragastric stent length is not obtained during the procedure, a CT should be performed to check for possible abnormalities such as stent migration, pneumoperitoneum, or fluid collection. If the intragastric stent length is no longer sufficient due to stent shortening or gastric movement (impending migration; Figure 14B,C), there is a risk of migration of the gastric end of the stent into the abdominal cavity. In such cases, immediate endoscopic reintervention using various technique such as Crisscross anchoring technique [88], Clip-flap technique [89], and Stent-in-stent technique [90] should be performed to prevent stent migration. Pneumoperitoneum or fluid collection with new-onset abdominal pain or fever suggests bile leak, and antibiotics should be continued. If melena or an unexpected drop in hemoglobin is seen, a contrast-enhanced CT is necessary. When bleeding from hepatic artery is suspected, angiography should be performed urgently. The results of the pooled analysis of early adverse events of EUS-HGS described in the Japanese clinical practice guidelines are summarized in Table 2.

## 7. Conclusions

This review describes the technical tips for safe and successful EUS-HGS, in particular the method using a covered SEMS for palliative drainage purposes. Recent advances and innovation in EUS systems, equipment, and methods have made EUS-HGS an easier and safer procedure, but the risk of serious adverse events such as stent migration and bile leak still remains. The techniques described in this article are all practical and should be readily available, especially for physicians who are just starting EUS-HGS. It is hoped that further advances in instrumentation will make EUS-HGS safer and more reliable.

## Figures and Tables

**Figure 1 jcm-11-01591-f001:**
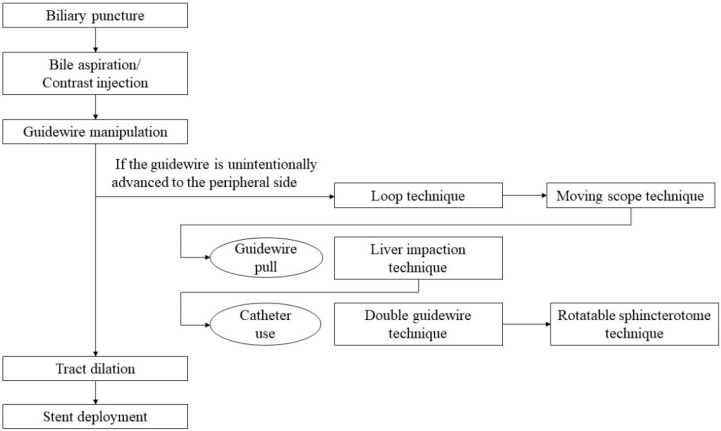
A flow diagram of step-by-step procedures in EUS-HGS.

**Figure 2 jcm-11-01591-f002:**
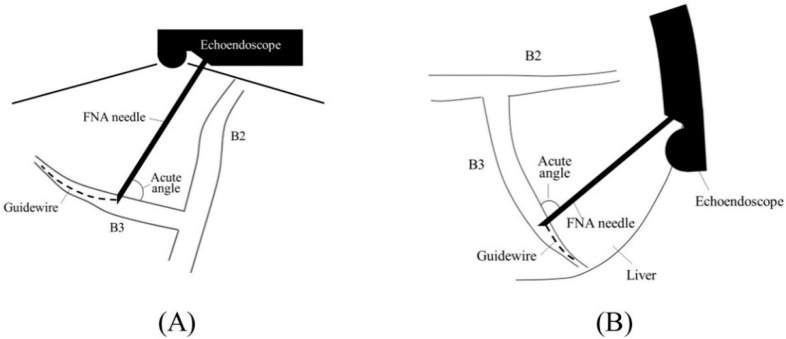
Too shallow echoendoscope position in B3 puncture. In a shallow scope position, the angle formed by a needle and the bile duct on the hilar side is often acute, and a guidewire can easily go to the peripheral side ((**A**); ultrasound image, (**B**); fluoroscopic image).

**Figure 3 jcm-11-01591-f003:**
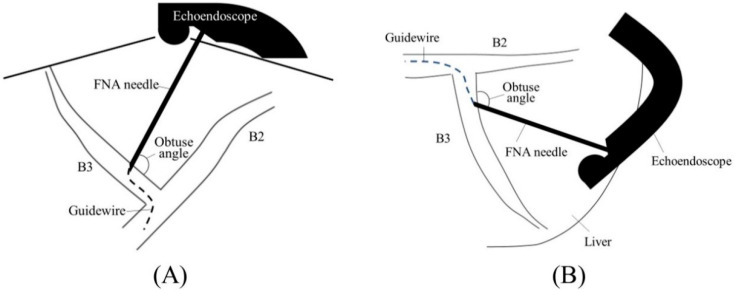
Optimal echoendoscope position in B3 puncture. Pushing a scope while turning the large wheel upward rotates the EUS image clockwise and makes the angle between a needle and the bile duct on the hilar side obtuse, making a guidewire manipulation toward the hilar region easy (**A**). Fluoroscopic image (**B**).

**Figure 4 jcm-11-01591-f004:**
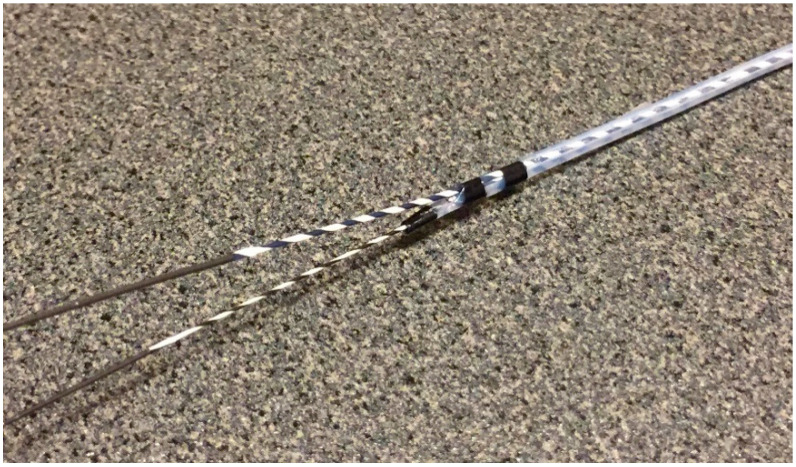
Uneven Double Lumen Cannula (Piolax Medical Device). The double lumen catheter allows a second 0.035 inch guidewire to be inserted adjacent to the first 0.025 inch guidewire. (Courtesy of Piolax Medical Device).

**Figure 5 jcm-11-01591-f005:**
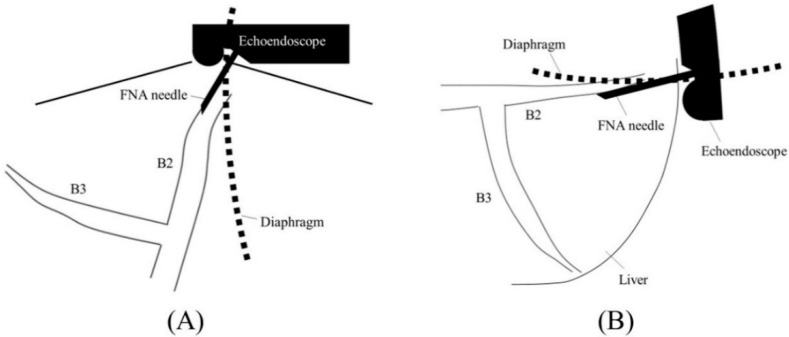
Too shallow echoendoscope position in B2 puncture. In a shallow scope position, a needle and B2 are parallel to each other, making puncture difficult and increasing the risk of transesophageal puncture ((**A**); ultrasound image, (**B**); fluoroscopic image).

**Figure 6 jcm-11-01591-f006:**
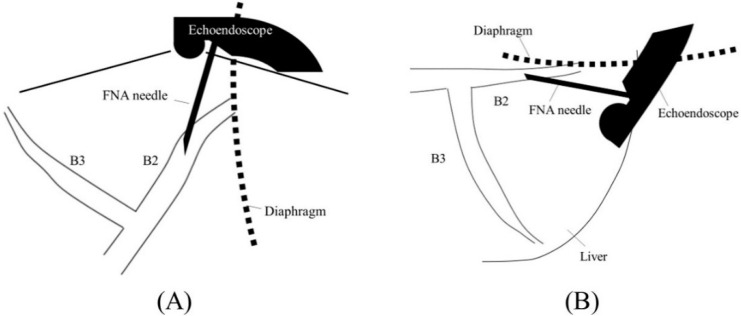
Optimal echoendoscope position in B2 puncture. Pushing a scope while turning the large wheel upward facilitates transgastric and reliable bile duct puncture ((**A**); ultrasound image, (**B**); fluoroscopic image).

**Figure 7 jcm-11-01591-f007:**
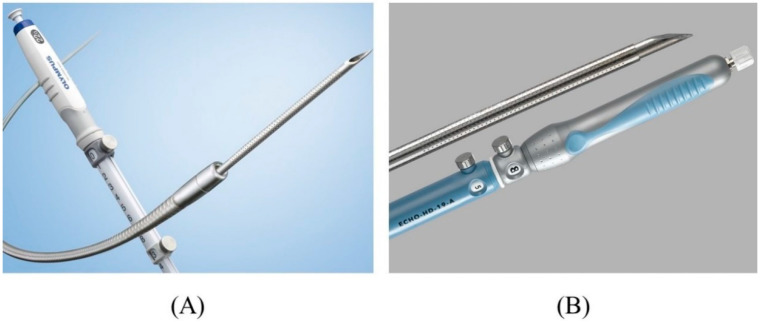
Needles suitable for EUS-HGS. EZ shot 3 plus (Olympus Medical Systems) has a nitinol needle with a coil sheath (Courtesy of Olympus Medical Systems) (**A**). EchoTip Access Needle (Cook Medical) has a sharp stylet and blunt-tipped needle (Courtesy of Cook Medical) (**B**).

**Figure 8 jcm-11-01591-f008:**
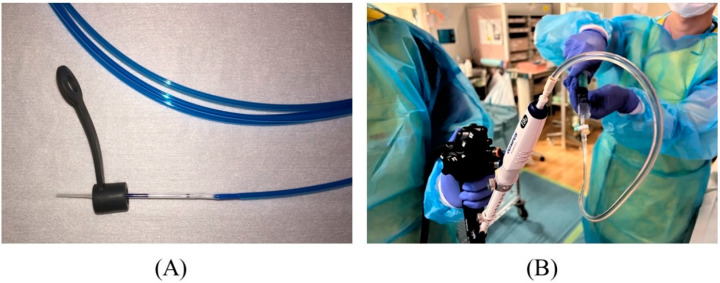
Preparation for puncture. The biopsy valve is attached to a dilation device (**A**). The needle stylet is removed, and a syringe filled with contrast medium is attached to the needle to pre-fill the lumen with contrast medium (**B**).

**Figure 9 jcm-11-01591-f009:**
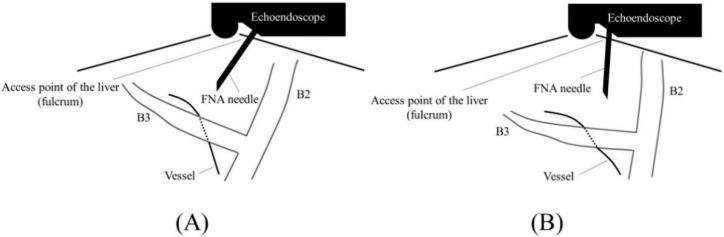
Changing a needle trajectory during biliary puncture. If a favorable biliary puncture line cannot be obtained due to the intervening vessels (**A**), pushing a scope after advancing a needle into the liver parenchyma to change the needle direction using the liver access point as a fulcrum (**B**).

**Figure 10 jcm-11-01591-f010:**
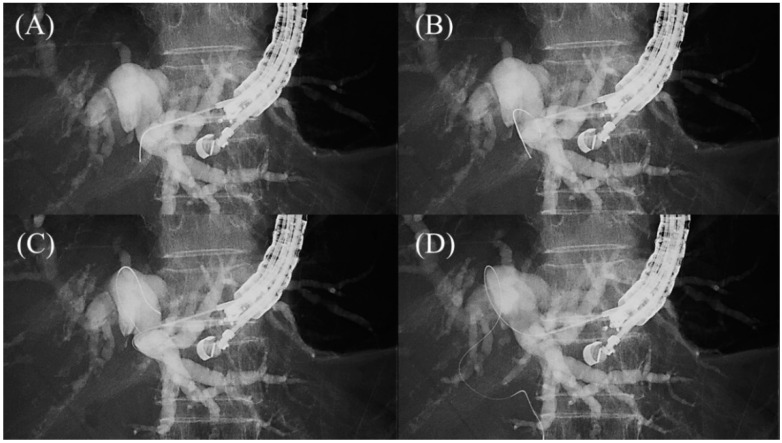
Loop technique for redirection of a guidewire. If a guidewire is unintentionally advanced to the peripheral side, push the guidewire with rotation. When the tip of the guidewire is caught on a lateral branch (**A**), the guidewire will bend and form a loop by pushing force (**B**). If the loop is facing the hilar region, the guidewire can be advanced to the hilum by pushing further (**C**,**D**).

**Figure 11 jcm-11-01591-f011:**
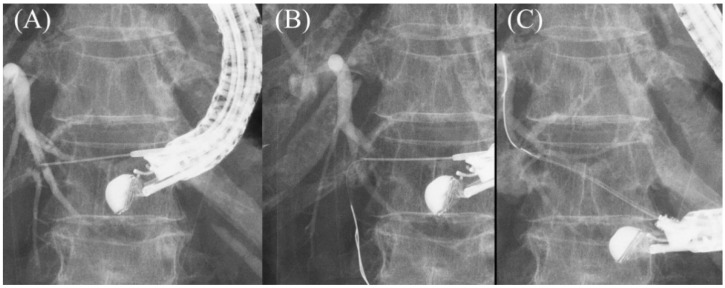
Moving scope technique for redirection of a guidewire. If a guidewire is unintentionally advanced to the peripheral side (**A**,**B**), push the scope while turning the large wheel upward to change the needle direction to the cranial side, allowing the guidewire to proceed toward the hilum (**C**).

**Figure 12 jcm-11-01591-f012:**
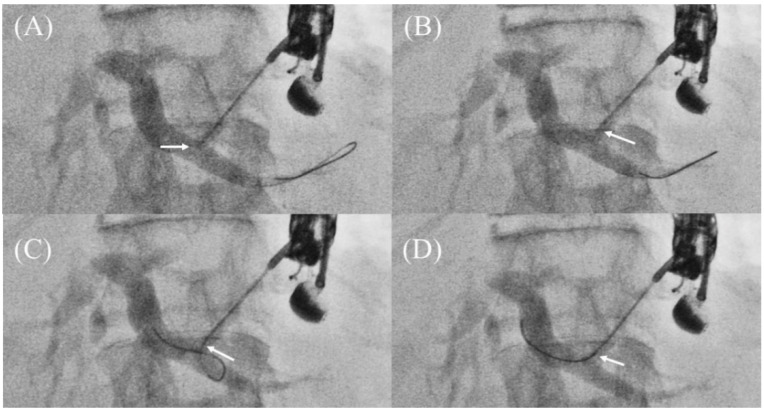
Liver impaction technique for redirection of a guidewire. If a guidewire is unintentionally advanced to the peripheral side (**A**), pull the needle tip slightly into the hepatic parenchyma (**B**). The guidewire can be pulled without shearing because the tip of the needle is covered by the hepatic parenchyma (**C**). The guidewire is successfully manipulated toward hilum (**D**). Arrows indicate the tip of the needle.

**Figure 13 jcm-11-01591-f013:**
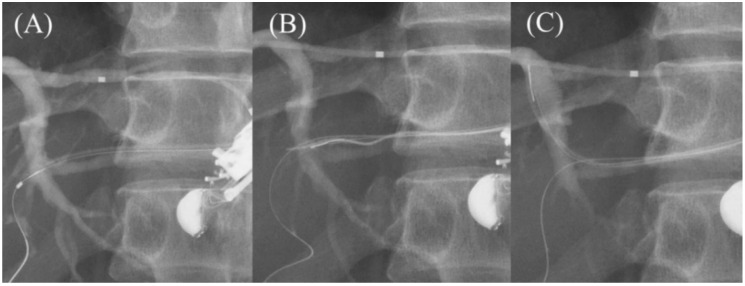
Redirection of a guidewire using a rotatable sphincterotome. In cases where the guidewire cannot be advanced toward hilum even using double guidewire technique (**A**), the guidewire is manipulated with a rotatable sphincterotome by rotating and bending the tip of the catheter while securing the bile duct with another guidewire (**B**). The catheter is successfully advanced toward the hilar region (**C**).

**Figure 14 jcm-11-01591-f014:**
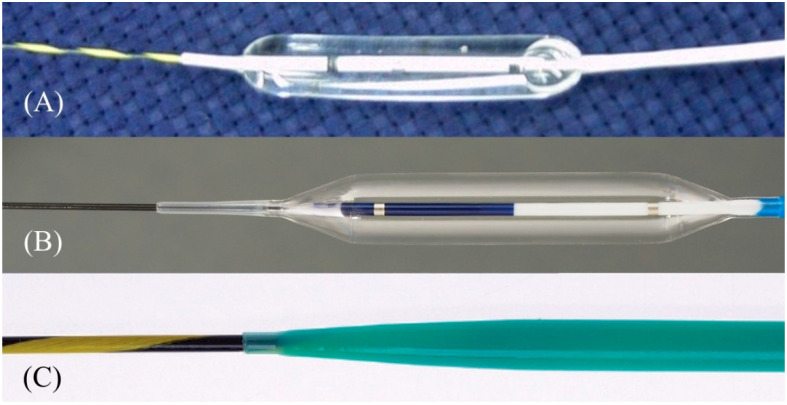
One-step mechanical dilation devices. Hurricane (Boston Scientific) is a balloon dilator with a rigid shaft and stylet (Courtesy of Boston Scientific) (**A**). REN (Kaneka Medics) is a balloon dilator with an ultra-tapered tip adapted to a 0.025 inch guidewire (Courtesy of Kaneka Medics) (**B**). ES dilator (Zeon Medical) is a bougie dilator with an ultra-tapered tip adapted to a 0.025 inch guidewire. (Courtesy of Zeon Medical) (**C**).

**Figure 15 jcm-11-01591-f015:**
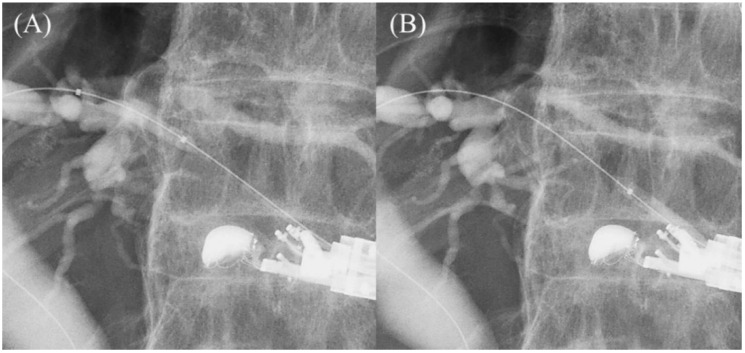
Segmental dilation method for prevention of bile leak during balloon dilation. A balloon catheter is pushed into the bile duct as deeply as possible when dilating the bile duct wall (**A**) and pulled into the scope channel as long as possible when dilating the gastric wall to prevent overlap of the two dilated areas (**B**). The hepatic parenchyma left un-dilated is thought to prevent bile leakage due to the tamponade effect.

**Figure 16 jcm-11-01591-f016:**
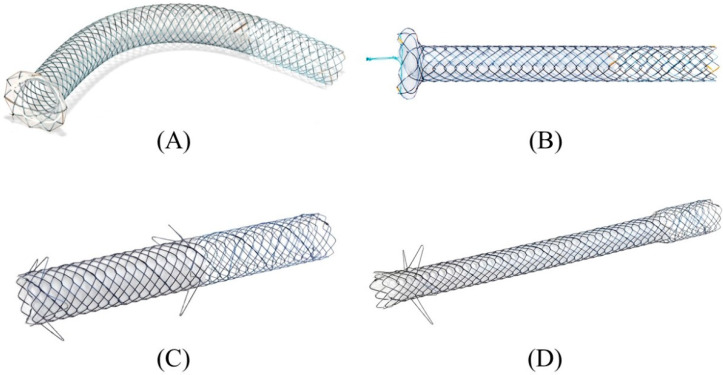
Partially covered SEMSs with anti-migration properties dedicated for EUS-HGS developed by Korean companies. GIOBOR stent (Taewoong medical) (**A**). HANARO stent BPD (M.I.Tech, Seoul, Korea) (**B**). Hybrid BONA stent (Standard Sci. Tech, Seoul, Korea) (**C**). DEUS (Standard Sci. Tech) (**D**). Courtesy of each company.

**Figure 17 jcm-11-01591-f017:**
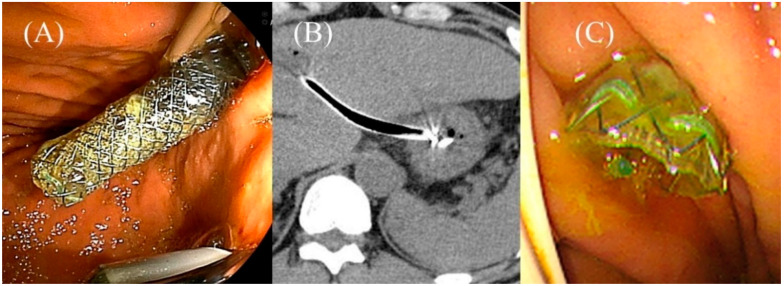
Impending delayed migration in Niti-S S-type stent (Taewoong Medical). A sufficient length of the gastric end of the stent is seen after the procedure (**A**). The next day’s CT shows that the intragastric stent length has shortened (**B**). Urgent endoscopy reveals impending migration of the gastric end of the stent (**C**).

**Figure 18 jcm-11-01591-f018:**
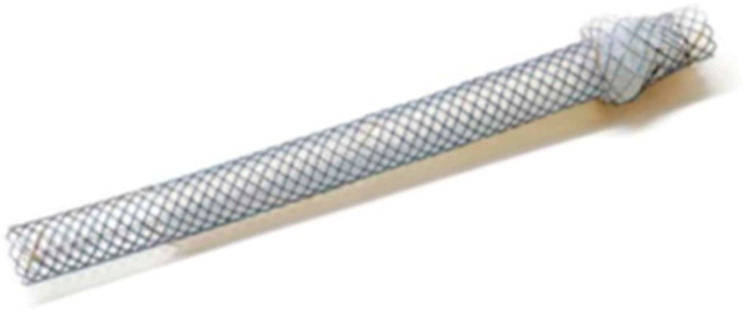
Spring Stopper Stent (Taewoong Medical), which has a spring-type stopper as an anti-migration system at the gastric end. (Courtesy of Taewoong Medical).

**Figure 19 jcm-11-01591-f019:**
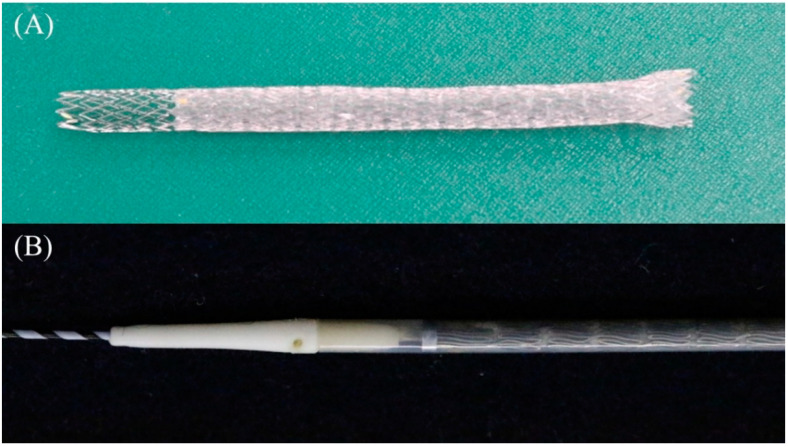
Covered BileRush Advance (Piolax Medical Device). The partially covered laser-cut stent of 8 × 120 mm in size with a 2 cm uncovered portion on the hepatic end (**A**). The slim introducer with a 7 Fr shaft and 2.4 Fr tip (**B**). (Courtesy of Piolax Medical Device).

**Figure 20 jcm-11-01591-f020:**
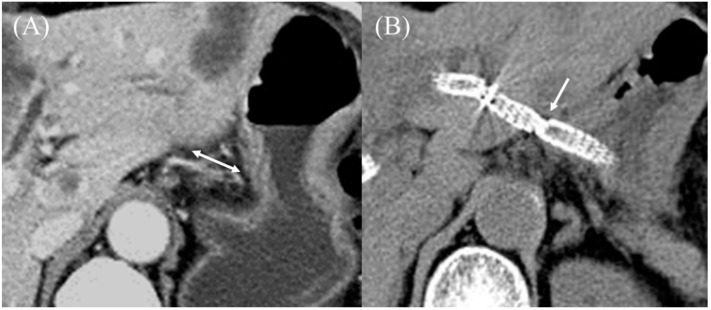
Endoscopic ultrasound-guided hepaticogastrostomy with a Covered BileRush Advance. Pre-procedure contrast-enhanced CT showed a long distance between the gastric body and left hepatic lobe (double arrow) (**A**). Post-procedure CT showed the Covered BileRush Advance fixed the gastric body near the left hepatic lobe by its jagged surface (arrow) (**B**).

**Figure 21 jcm-11-01591-f021:**
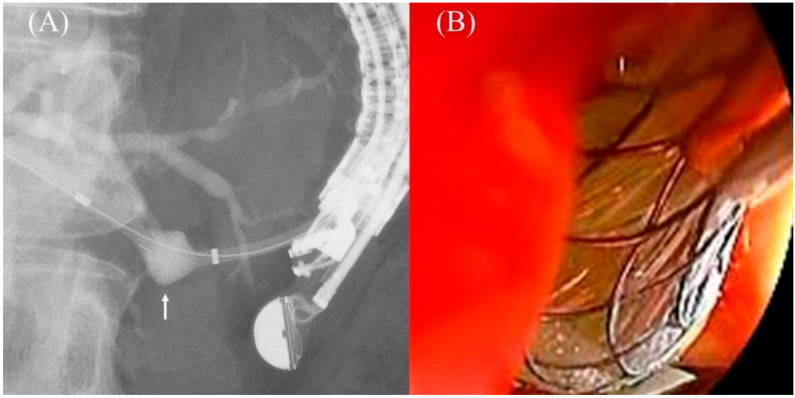
Intra-channel (conduit) release method. After pulling the introducer until 1 to 2 cm release inside the channel, push the expanded part of the stent (arrow) to strongly press the gastric wall for keeping the stomach and liver close together (**A**). Stent deployment across the gastric wall can be directly confirmed by endoscopic view (**B**).

**Table 1 jcm-11-01591-t001:** Specifications of endoscopic ultrasound systems.

		EG-580UT (Fujifilm)	GF-UCT260 (Olympus)	EG38-J10UT (Pentax)
Endoscopic Functions	Viewing direction	Forward oblique viewing 40°	Forward oblique viewing 55°	Forward oblique viewing 45°
	Observation range	3–100 mm	3–100 mm	3–100 mm
	Field of view	140°	100°	120°
	Distal end diameter	13.9 mm	14.6 mm	14.3 mm
	Insertion tube diameter	12.4 mm	12.6 mm	12.8 mm
	Bending capacity up/down	150°/150°	130°/90°	160°/130°
	Bending capacity left/right	120°/120°	90°/90°	120°/120°
	Working channel diameter	3.8 mm	3.7 mm	4.0 mm
	Working length	1250 mm	1250 mm	1250 mm
	Total length	1550 mm	1555 mm	1566 mm
Ultrasound Functions	Dedicated processor	SU-1	EU-ME2	None
	Sound method	Electronic curved linear array	Electronic curved linear array	Electronic curved linear array
	Scanning area	150°	180°	150°
	Frequency	5–12 MHz	5–12 MHz	5–13 MHz
	Scanning mode	B-Mode, M-Mode, Color Doppler, Power Doppler, Pulse Doppler	B-Mode, Color Flow Mode, Power Flow Mode	Depends on ultrasoundplatforms(ARIETTA series)

**Table 2 jcm-11-01591-t002:** Adverse events of EUS-HGS.

Adverse Event	Incidence
Overall	18.2%
Bleeding	3.7%
Bile leak	2.8%
Biloma	2.6%
Stent migration	1.6%
Stent misplacement	1.2%
Intrahepatic hematoma	1.2%
Sepsis	1.2%

## Data Availability

Data sharing not applicable.

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
