# Peer review of "Practical Tips for Safe and Successful Endoscopic Ultrasound-Guided Hepaticogastrostomy: A State-of-the-Art Technical Review"

_jcm, 2022, doi:10.3390/jcm11061591_

Round 1

Reviewer 1 Report

Thank you for your responses and the changes made on the manuscript. 

I'm sure that this review will help our colleagues to improve their technique of hepaticogastrostomy.

Reviewer 2 Report

Good Work

Reviewer 3 Report

This is a very interesting, informative, fluent and easy to read guidance on endoscopic ultrasound-guided hepaticogastrostomy, with well chosen illustrations. The author's hard work can be seen. The relevance of this paper compared to the other published technical tips and guidelines is correctly emphasized. The authors revised the manuscript to integrate discussions on systems and devices available worldwide, not only in Japan. 

This manuscript is a resubmission of an earlier submission. The following is a list of the peer review reports and author responses from that submission.

Round 1

Reviewer 1 Report

This is a well written, comprehensive and didactic review on technical considerations about hepaticogastrostomy.

I appreciated the schemas with both endoscopic and fluoroscopic view to help clinicians to best represent how to improve their procedures. 

I only have a few comments to improve the paper.

1) In the introduction, you could remove some of the references which do not add value (10 references only for the first sentence is a little bit to much).

2) since only one retrospective single center sudy focused on bleeding risk using a cystotome versus a mechanical dilatator, I would avoid to advice to use the diathermic dilation only in last resort. Diathermic dilation is routinely used in Europe and data available suggests similar complication rates than eastern countries. Anyway, the futur is ultra-thin stents catheter without requiring any dilation as you discussed in the paper. 

3) I don't believe that systematic CT-scan the day after procedure is systematically needed. Laboratory test, and more important, physical examination allows to suspect adverse events and do a CT only if stent misdeployment or migration is suspected. If the deployment of the stent was satisfying during procedure with sufficient intragastric lenght, a reintervention based on CT findings without clinical pain or sepsis could be contre-productive. Better is the enemy of 'good enough' !

Reviewer 2 Report

I found the work really interesting, well detailed and very useful. It would be useful to add an initial figure with a diagram (such as a flow-chart) with the graphic description of the step by step procedure.

It would also be important to add a final table with the list of the most important complications and their percentages from the most recent literature.

It is necessary to revise the bibliography: there are some incomplete citations; e.g. 54, 88,90 ...For the rest it is a really good job

Reviewer 3 Report

This is a very interesting, informative, fluent and easy to read guidance on endoscopic ultrasound-guided hepaticogastrostomy, with well chosen illustrations. The author's hard work can be seen.

However, there is an important problem I wish to clarify before acceptance for publishing.

There are currently some guidelines (https://onlinelibrary.wiley.com/doi/10.1002/jhbp.631, https://www.ncbi.nlm.nih.gov/pmc/articles/PMC6896433/) and technical reviews that provide practical tips for EUS-HGS ( https://www.gutnliver.org/journal/view.html?uid=1746&vmd=Full, https://www.ncbi.nlm.nih.gov/pmc/articles/PMC4823244/).

1) A discussion in the introduction about similar technical reviews with highlights on why this review is needed and what does it bring new compared to them, would be welcome.
2) The authors only talk about various types of systems and instruments produced or available in Japan. Does this paper aims mainly to address japanese particularities and readership? If yes, then the title or the introduction should clarify this scope. But if not, adding examples of systems or instruments produced in other important centers from the world would be much more informative for the general readership (for example, table 1 could be expanded; and several other examples throughout the paper)